# Protocol for the development of a core outcome set for cauda equina syndrome: systematic literature review, qualitative interviews, Delphi survey and consensus meeting

Nisaharan Srikandarajah,[1] Adam J Noble,[2] Martin Wilby,[3] Simon Clark,[3] Paula R Williamson,[1] Anthony Guy Marson[1]

[1]Institute of Translational Medicine, University of Liverpool, Liverpool, UK
[2]Institute of Psychology Health and Society, University of Liverpool, Liverpool, UK
[3]Department of Neurosurgery, The Walton Centre NHS Foundation Trust, Liverpool, UK

**Correspondence to**
Mr Nisaharan Srikandarajah; nishsri09@gmail.com

## ABSTRACT

**Introduction** Cauda equina syndrome (CES) is a serious neurological condition most commonly due to compression of the lumbosacral nerve roots, which can result in significant disability. The evidence for acute intervention in CES is mainly from retrospective studies. There is heterogeneity in the outcomes chosen for analysis in these studies, which makes it difficult to synthesise the data across studies. This study will develop a core outcome set for use in future studies of CES, engaging with key stakeholders and using transparent methodology. This will help ensure that relevant outcomes are used in future and will facilitate attempts to summarise data across studies in systematic reviews.

**Methods and analysis** A systematic literature review will document all the outcomes for CES after surgery mentioned in the literature. The qualitative interviews with patients with CES will be semistructured, audio recorded, transcribed and thematically analysed with the use of NVivo V.10 to identify outcomes and determine the themes described. The outcomes from the literature review and patient interviews will be combined and prioritised to determine what the most important outcomes are in CES research studies to patients and healthcare professionals. The prioritisation will be done through a two-round iterative Delphi survey and a consensus meeting. This process will decide the core outcome set for patients with CES.

**Ethics and dissemination** REC and HRA approval was obtained on the 6/12/16 for the qualitative interviews from South Central—Hampshire A REC. REC reference 16/SC/0587. REC and HRA approval was obtained on 26/3/18 for the Delphi process and consensus meeting from North West—Greater Manchester Central REC. REC reference was 18/NW/0022. The final core outcome set will be published and freely available.

**Trial registration number** This study is registered with the Core Outcome Measures in Effectiveness Trials database as study 824.

## Strengths and limitations of this study

► A systematic literature review following Preferred Reporting Items for Systematic Reviews and Meta-Analyses guidelines will identify outcomes in the existing literature for cauda equina syndrome (CES).
► Semistructured qualitative interviews using a sampling frame to select a varied sample of patients with CES will identify outcomes important to them.
► The consensus process of an international online Delphi survey and an international face to face consensus meeting will involve patients and healthcare professionals.
► A core outcome set will allow future CES research studies to use outcomes relevant to key stakeholders and allow synthesis of data in CES.
► The outcomes that constitute the core outcome set will be reported. 'How' these outcomes are measured will not be determined in this study and requires further work.

beneath the conus medullaris resulting in sensorimotor deficits of the lower limbs and sphincter dysfunction. Symptoms and signs include low back pain, unilateral or bilateral sciatica, saddle anaesthesia and motor weakness of the lower extremities with bladder and/or bowel dysfunction.[1 2] The most common cause of CES is a herniated lumbar disc, and represents 2% of all herniated lumbar discs. CES has an incidence of 2 per 100 000 in England and is an indication for emergency decompression surgery.[3–5] Other less common aetiologies include spinal stenosis, spinal tumours, haematomas, fractures and infections.[2] The National Spinal Task Force showed that there are 981 operations done each year for CES in the UK from 2010 to 2011.[6] Surgical intervention for CES is not a rare procedure, and the economic burden of severe disability is a worrying

## INTRODUCTION

Cauda equina syndrome (CES) is due to dysfunction of the lumbosacral nerve roots

unknown for both patient quality of life and development of appropriate health services.

The evidence for acute intervention in CES is mainly from retrospective studies.[7 8] The importance of categorising CES into CES incomplete (CESI) and CES complete with urinary retention (CESR) has been highlighted in the literature.[4] CESR describes painless urinary retention with overflow incontinence and complete perianal sensory loss. When the patient had CESI, the symptoms include urinary issues of neurogenic origin including loss of desire to void, altered urinary sensation and hesitancy with partial saddle anaesthesia.

It is documented in the literature that timely operative decompression for CES secondary to herniated lumbar disc can lead to improved outcomes in patients.[7–9] In fact, delay or missed diagnosis of this condition incurs heavy litigation costs to the NHS at £336000 (US $549 427) per case on average[10] as reported to the Medical Defence Union in the UK.

### Rationale for the development of a COS

An 'outcome' in relation to clinical research studies is defined to be a measurement or observation used to capture and assess the effect of treatment such as assessment of the side effects (risk) or effectiveness (benefits).[11]

Before the systematic literature review a scoping review was undertaken.[12] It was identified that there were no randomised controlled trials, many retrospective observational studies and few prospective studies reporting the clinical outcome of patients with CES. There is heterogeneity and inconsistency in the outcomes reported in the literature for CES. The outcomes reported in the literature have not been independently validated as important to key stakeholders.

There is no defined core outcome set (COS) in CES currently, and this protocol will describe the methods of how to develop it. A COS defines the minimum outcomes that should be consistently measured and reported in clinical trials in a specific area of healthcare.[13] With this there will be greater reporting consistency and a reduction in outcome reporting bias in healthcare studies contributing to systematic reviews and meta-analysis[14] that can lead to informed healthcare decisions.

Initially, a systematic literature review and qualitative patient interviews will be conducted to document the outcomes for patients with CES after surgery. These outcomes will be combined and prioritised through two rounds of a Delphi process with key stakeholders and a consensus meeting to decide the COS. The COS would be published and used for future research studies and improving outcome reporting in CES.

The development of COSs has been done successfully in rheumatology with the Outcomes Measures in Rheumatoid Arthritis Clinical Trials group. This international collaboration was developed in the early 1990s involving patients in the development of COSs and has improved consistency of reported trials in this specialty.[14 15] The Core Outcome Measures in Effectiveness Trials (COMET)

initiative advocates the involvement of patients and currently holds a database of on-going COS developers[16] to minimise duplication and foster health service user engagement.[13 17]

### Scope of the COS

We aim to identify 'what' outcomes of patients with CES are of concern to key stakeholders using transparent methodology. We are not intending to consider how these outcomes should be measured. The 11 minimum Core Outcome Set Standards for Development (COS-STAD) recommendations are addressed in this protocol[18] (table 1).

### Registration

The study is registered on the COMET database as study 824 (http://www.comet-initiative.org/studies/details/824?result=true).

### METHODS AND ANALYSIS

Development of the COS will be developed in four phases with their estimated time frames highlighted in the overall study timeline (figure 1). Timeframes includes the estimated duration for ethical approval, study recruitment and analysis.

### Phase 1: systematic literature review
#### Research question

What outcomes are reported in the medical literature after surgery for CES?

### Summary

The aim of the systematic literature review was to summarise the reporting standards of the clinical outcomes after surgery in patients with CES following the Preferred Reporting Items for Systematic Reviews and Meta-Analyses guidelines.[19] Most CES cases are due to lumbar disc herniation,[20] which requires urgent surgical intervention. Study inclusion was limited to articles with patients who were surgically managed and whose outcomes were recorded.

The systematic literature review summarised the outcomes that had been mentioned in the literature and categorised them into a known taxonomy.[21] About 1873 articles were identified through the search strategy of which 61 met the inclusion criteria. Inclusion criteria specified details regarding the study design, diagnosis, procedure, publication date, language and the patient age. Where 737 outcomes were reported verbatim in the 61 included articles. These were then categorised to 20 higher order groupings called 'outcome domains.' The most commonly reported outcomes were bladder function (70.5%), motor function (63.9%) and sensation (50.8%). There was significant variation in the terms used for each outcome for example, bladder function outcome domain had 141 different terms. Significant heterogeneity was evident in the outcomes reported in

**Table 1** Core Outcome Set Standards for Development recommendations

| Domain | Standard number | Methodology | Notes |
|---|---|---|---|
| Scope specification | 1 | The research or practice setting in which the COS is to be applied. | Research studies that will inform clinical decision making. |
| | 2 | The health condition(s) covered by the COS. | All severities of cauda equina syndrome (CES). |
| | 3 | The population(s) covered by the COS. | Human adults aged 18 or above. |
| | 4 | The intervention(s) covered by the COS. | Clinical management of CES including surgery. |
| Stakeholders involved | 5 | Those who will use the COS in research. | Clinical trialists in CES are healthcare professionals who manage patients with CES. They are included in standard 6. |
| | 6 | Healthcare professionals with experience of patients with the condition. | This will include clinicians, experts and healthcare professionals involved in CES management. |
| | 7 | Patients with the condition or their representatives. | Patients with a diagnosis of CES will be included.[45] |
| Consensus Process | 8 | The initial list of outcomes considered both healthcare professionals and patients views. | Systematic literature review[22] considered healthcare professional views. Qualitative interviews considered patient views. |
| | 9 | A scoring process and consensus definition were described a priori. | Described in the 'Scoring' and 'Analysis' section of this protocol. |
| | 10 | Criteria for including/dropping/adding outcomes were described a priori. | Described in the 'Analysis' section of this protocol. |
| | 11 | Care was taken to avoid ambiguity of language used in the list of outcomes. | Plain language and clinical explanations available. These will be pilot tested with patients and healthcare professionals. |

CES, cauda equina syndrome; COS, core outcome set.

CES research studies. This highlighted a need for a COS in CES to be developed.[22]

### Phase 2: qualitative interviews

#### Research question

What outcomes have CES patients experienced after surgery and how do they feel about the management before and after surgery?

### METHOD

The objectives of the qualitative interviews with patients with CES are:

► To explore the patient experience of living with CES.
► To document what the patient describes as the most important outcomes they are experiencing.
► To determine what service improvements can be made to improve CES management and aftercare.
► To determine who should be the key stakeholders in the Delphi survey.
► Identify appropriate language to use for the Delphi survey.[23]

These interviews will be documented with audio recorded transcripts. The list of all potential outcomes from the systematic review and qualitative interviews will be placed into outcome domains by the research team to avoid repetition by qualitative method of content analysis.[24] The qualitative interviews will be piloted with two patients with CES to establish if the interview structure and technique is clear, understandable and capable of answering the research questions. This would recognise any corrections that need to be made to the interview structure or technique. Inclusion and exclusion criteria are shown in table 2.

### Participant selection

Adult patients for the qualitative interviews will be selected from those coded as having a diagnosis of CES in the medical records. There is an existing database of patients with CES who have been operated on and followed up by consultants, registrars or nurse specialists depending on the next available clinic. Adult patients will be 18 years or older who have had spinal surgery to remove the compressive lesion at a single tertiary NHS institution over the past 10 years. The qualitative interviews will capture short and long-term outcomes that are deemed important to them. Duration of the recorded outcomes will be calculated since the initial operation for CES.

Stratified purposive sampling[25] was chosen in which the aim is to select groups that display variation in particular characteristics so the subgroups can then be compared. Characteristics known to have an impact on the outcomes being investigated have been identified—severity of CES (CESI or CESR)[4] then there is a subgroup about which little is known and whose circumstances and views need to be explored; short (≤2 years) or long term (>2 years and ≤10 years) since the operation (see table 3). This

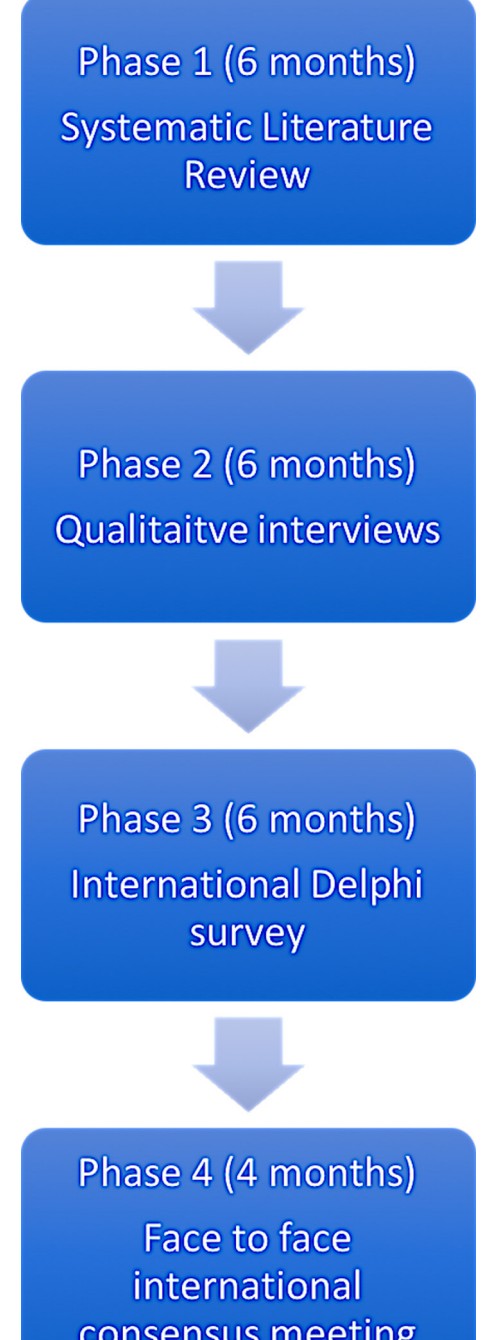

**Figure 1** The overall study timeline.

**Table 2** Inclusion and exclusion criteria for qualitative interviews

| Inclusion criteria | Exclusion criteria |
| --- | --- |
| Adult patients | Adults unable to consent for research |
| Diagnosis of cauda equina syndrome (CES) | |
| Patient underwent a surgical procedure for CES | |
| Less than 10 years since the surgical procedure | |
| Ability to converse in English and to consent for research | |

as long travel distance from institution, not interested in participating it is anticipated that up to 10 patients may reply from each category, which would produce up to 40 patients in total. Options will be given to be interviewed at home, via electronic media (Skype), over the phone or to attend the hospital in person. After informed consent, patients will be interviewed until 'data saturation' is reached. The research team will decide when data saturation is reached. Data saturation is the point where increasing the sample size no longer contributes to new evidence[26] moreover even large qualitative studies do not interview more than 50 people.[27] Additional patients will be interviewed in the subcategories if one group has a better response rate until data saturation is achieved.

Sticking rigidly to a sample frame could be counter intuitive as one patient can be data rich during the interview as opposed to interviewing five patients where data is not rich. The aim is to collect rich data to allow in depth analysis.[26] So, although the sampling frame may serve as a guide it will not be used to start restricting participants especially at the initial stages of doing the qualitative interviews until data saturation is achieved.

An information leaflet and stamped addressed envelope to return the response slip will be sent to participants with a consent form. Patients will have 3 weeks to 'opt-out' of the study by returning a response slip, through email or telephone with the research team. After this, the participants will receive a phone call from the research team to confirm interest for participating in the study, to answer any further questions and to arrange a time and location for the interview.

will produce four subcategories to populate. This is to prevent potential bias you may get from having many patients who presented with a severe clinical picture and poor outcomes being more forthcoming and vocal. All subcategories for the sampling frame will be deemed a priority. Half the participants would ideally be male and half would be female.

There is an existing database of 200 patients with contact details and clinical details of presentation and management, which will be updated up to the current date to exclude patients who are deceased. This should produce 50 patients per category. Due to reasons such

**Table 3** Sampling frame with suggested quotas

| | Cauda equina syndrome incomplete | Cauda equina syndrome with retention |
| --- | --- | --- |
| Short term since the operation (≤2 years) | 10 participants | 10 participants |
| Long term since the operation (>2 years and ≤10 years) | 10 participants | 10 participants |

## Interview format and analysis

A semistructured interview format will be used as per our topic guide (online supplementary file 1). Qualitative semistructured interviews were chosen over question-naires and focus groups as it was believed that patient opinions over sensitive subject matter such as bowel, bladder and sexual function would be better elicited in a private one-to-one interview and they were less likely to inhibit their contribution.[26] In addition, one-to-one interviews are more accessible for potential participants and for patients with mobility restrictions.

Informed consent will be obtained prior to the interview where anonymity and confidentiality will be expressed. The consent will also request the patient's permission for their general practitioner (GP) to be informed of their involvement in the study. This is so that if there is any distress during the patient interviews, which requires medical management they can be referred to their GP. Open-ended non-leading questions on their diagnosis, management postoperatively in hospital and management in the community will be asked allowing the participant to describe their experiences without unnecessary interruption.[27] Discussion will be directed towards outcomes of importance to the patient as seen in the topic guide. The interviewer will not discuss their own opinions about CES, and if these are asked they will be answered at the end of the interview session. Reflexivity is an important concept during qualitative research for striving towards objectivity and neutrality,[26] and the analysis of the interviews will consider if bias from the interviewer's own beliefs may have crept in. It is anticipated that the interview will last for 45 min to an hour at each sitting to prevent the participant feeling fatigued. The same interviewer (NS) will be used for all the patient interviews. All interviewees will be made aware that the interviewer is a doctor not involved in their on-going care. A sample of the transcripts will be reviewed by a supervisor not involved in the qualitative interviews to confirm that they were undertaken in a satisfactory manner.

Initially, the transcripts will be reviewed to start identifying which outcomes are important to the patients by labelling the data using NVivo qualitative data analysis software V.10. A pragmatic approach will be taken by using thematic analysis as per the Braun and Clarke method.[28] It is a pattern-based qualitative method like grounded theory[29] and interpretative phenomenological analysis[30] but is not linked to a specific theoretical framework. This method will allow summarisation of the key outcomes of each individual transcript and overall themes while retaining the context and language in which it was expressed.[26] The qualitative interviews will be reported as outlined by the Consolidated criteria for Reporting Qualitative research (COREQ); a 32-item checklist.[31]

## Phase 3: the Delphi survey

The outcomes from the systematic literature review and qualitative interviews will create a long list.[11] This will be condensed by grouping similar outcomes into domains and conforming with the taxonomy used in the systematic literature review.[21 22] This will be reviewed and agreed by the study team and pilot tested with the key stakeholders before the Delphi survey is distributed.

## Research question

Which outcomes do patients and healthcare professionals think should be included in a COS for patients with CES?

## METHOD

All patients with CES will be invited to participate in the Delphi survey regardless of whether they had had surgery or not. Although there are a minority of participants in the category of non-operative management of CES[32] it was decided by the study team that including them will be an opportunity to consider their input and maximise recruitment. The Delphi will be done by healthcare professionals and patients.

To achieve a priority list, we will use the 'modified' Delphi method[33] as opposed to the 'traditional' Delphi method.[34] Traditionally in a Delphi survey patients are asked open questions in the first round of the Delphi and the answers would constitute the outcomes rated in the second round. In the 'modified' Delphi, which will be used in this study, rating the outcomes will take place over two rounds. A list of outcomes previously attained from the systematic literature review and qualitative interviews will be presented in the first round of the Delphi.[33] Patients can also suggest outcomes that have not been mentioned in the first round, but these will not be scored. They will be considered for inclusion into the second round of the Delphi if, as judged by the CES study team, the outcome does not reflect or is not similar to another outcome already listed. The CES study team includes a patient representative.

The level of anonymity will be 'fully anonymised'[35] so participants do not know the identities of other individuals in the group and they will not know the specific answers other individuals give. In round 2 of the Delphi, participants will know the group responses from the patient group and the healthcare professional group. Individual participants can decide to keep their original rating or to change their rating in the next round. This will lead to the group converging on a consensus opinion over the course of these two rounds.[35]

## Inclusion criteria

Participants will be recruited from two key stakeholder groups: patients and healthcare professionals. All participants will be adults over 18 years of age and able to complete an online survey in the English language.

### Patients

Participants who have had an operation for CES.

### Healthcare professionals

All members of the clinical team involved in directly caring for a patient with CES such as:

- Spinal surgeons
- Spinal specialist nurses
- Neurorehabilitation doctors

## Sampling and recruitment
### Patients
At the main site, the clinical care team have a pre-existing database of patients with CES they have clinically managed. The clinical care team will send an invitation letter to the home address of these patients. There will be no follow-up calls or further correspondence. It is the patient's decision if they wish to be involved and the invitation will contain details of the website address patients can access if they wish to find out more details regarding the study. Online patient groups for CES will be contacted internationally. A named contact for each group will act as the liaison member to circulate the participant invitation email and poster. This may include the patient groups sharing the recruitment details on social media.

### Healthcare professionals
The main study site has spinal multidisciplinary team meetings held weekly. The coordinator has a preset mailing list that goes to healthcare professionals involved in the meeting. This will be used to send the participant invitation email. The membership of national and international associations will be contacted and invited to participate. They include different healthcare professionals in their membership categories. Some examples are listed below:

- Society of British Neurological Surgeons
- British Association of Spine Surgeons
- World Federation of Neurorehabilitation
- Spinal Injuries Association

Known contacts of the CES study group will be contacted and invited to participate. Snowballing sampling will be used to increase the sample size. The participant invitation email/letter will be the first contact for healthcare professionals and patients, which is a short introduction and summary of the study. If they are interested further the participant can proceed to the registration website for further details and obtain a copy of the participant information leaflet.

### Sample size
There are no strict recommendations for the number of participants (patients and healthcare professionals) required in a Delphi study to gain consensus.[35] In general, having more participants will increase the reliability of the group judgement.[36] A pragmatic approach to sample size will be taken and all individuals who meet the inclusion criteria as identified above will be invited to participate. The recruitment phase will be 2 months before the first round of the Delphi survey is released. Documentation of the organisations who distribute the Delphi invitation from each stakeholder group will be recorded. No further participants will be invited after the first round of the Delphi.

## Consent
Consent will be implicit by the participant (patients and healthcare professionals) registering their name and email address to take part in the Delphi survey via the website.

## Questionnaires
The questionnaire is constructed and delivered in an online format using the DelphiManager software developed by the COMET initiative. Before starting the questionnaire, the participant will be asked to clarify which of the two stakeholder groups they belong to. For each stakeholder group, specific information will be collected:

- Patients—age, gender, location, surgery for CES—yes/no, years since surgery for CES, employed— full time/employed part time/unemployed.
- Healthcare professionals—practising field (spinal surgeon, specialist nurse, neurorehabilitation etc), years in practice, location, gender.

Following confirmation of their eligibility to participate in the study, participants will be sent an on-line link to access the first round of the Delphi process. Instructions of how to complete the questionnaire will be included at the beginning of each round. Only participants who respond to the first round of the Delphi will be invited to participate in second round taking the assumption that if they had not participated in the first round they would be unwilling to participate in the second round. Data will be collected over at least a 4-week period for each round of the Delphi process. Participants who have not completed the survey will be sent reminders via email when they have 2 weeks, 1 week and 48 hours remaining for completion of the survey. Participants who have not completed the questionnaire within 4 weeks of the start will be deemed not to have completed that round of the Delphi. The language used by patients in the qualitative interviews will be used to help term the outcomes for the Delphi. Plain language summaries by the COMET Patient Participation, Involvement and Engagement group was used to develop the Delphi information sheet. The Delphi will be piloted with two participants from each stakeholder group to highlight any issues with understanding or validity.

## Scoring
For an outcome to be included in the COS, there must be a majority agreement of the critical importance of the outcome and minority agreement that the outcome is not important.[37] This is in par with the Grading of Recommendations Assessment, Development and Evaluation working group recommendations (http://www.gradeworking-group.org;.[38 39] At the beginning of the Delphi, participants will be reminded the importance of completing the entire Delphi process. Round one of the Delphi study, we will ask participants to rate each outcome using a nine point Likert scale. This scoring system was chosen after previous studies and expert databases showed it differentiates the most between questionnaire items.[16 35] From 7 to 9 indicates critical importance. Where 4 to 6 represents

**Table 4** Definitions of a consensus

| Classification of consensus | Description | Definition |
|---|---|---|
| In | Consensus that outcome should be included in the core outcome set. | 70% or more participants scoring as 7 to 9 AND <15% participants scoring as 1 to 3 in both stakeholder groups. |
| Out | Consensus that outcome should not be included in the core outcome set. | 50% or less participants scoring 7 to 9 in both stakeholder groups. |
| No consensus | Uncertainty about importance of outcome. | Anything else. |

outcomes that are important but not critical while 1 to 3 are deemed to be of limited importance. All outcomes will be carried through to second round with anonymised feedback of first round scores from the patient group and from the healthcare professional group displayed for each outcome. The feedback will show the cumulated scores from each stakeholder group for each outcome, and the participant will be asked to rate the outcomes again using the same nine point Likert scale. If they change their score on the second round they will have the opportunity to explain their reasoning for this. Outcomes which have been suggested in round 1 by the participants and deemed appropriate by the study group will then be entered in for rating in the second round by key stakeholders. After the final Delphi round, there will be a list developed from all stakeholder groups, which will be submitted to a face to face consensus meeting of key stakeholders to discuss what outcomes that should be finally included in the COS. All participants who had completed both rounds of the Delphi survey will be eligible for invitation to the consensus meeting. A trained independent facilitator would chair this meeting.

## Analysis

Consensus that an outcome should be included in the COS is defined as 70% or more scoring it as 7 to 9 and fewer than 15% scoring it as 1 to 3, which is has been seen to be successful with the development of other COSs[40 41] (table 4). This will be done for each stakeholder group. Results at the multiple rounds of the Delphi process and consensus meeting will be documented to include the number of participants invited, number completing the section, measure of each group response to an outcome leading to a comprehensive list of all outcomes that should be included in the COS CES.

## Attrition

It is expected that some participants will drop out after each round of the Delphi. Each participant will be given a unique participant number when they complete the first round of the Delphi, which will allow calculation of the attrition rates between rounds. This will allow identification of participants who have completed all rounds and see if there is any difference bias between those participants who complete the process. Mean round 1 scores for the participants who completed round 1 and round 2 will be compared with those that dropped out after round 1.

## Phase 4: consensus meeting

All participants registering for the Delphi survey will be asked if they would be happy to attend a face to face consensus meeting involving patients and healthcare professionals. They will need to complete both rounds of the Delphi survey to be eligible to attend. This would be set up as a tick box on the registration page for the online Delphi.

Forty participants will be invited to the consensus meeting. This will include 20 healthcare professionals and 20 patients. Out of the 40 participants; 30 will be from the UK and 10 will be international. Standard travel expenses and hotel accommodation will be reimbursed or provided. Ten of the participants at the consensus meeting will be invited before the Delphi survey is released to attend the consensus meeting but on the premise, that both rounds of the Delphi are completed. This is to make sure there is representation at the consensus meeting from key stakeholder organisations closely involved with patients with CES, research or management. Thirty participants at the consensus meeting will be those who have completed both rounds of the Delphi and ticked their interest to attend the consensus meeting during registration.

In the development of a breast reconstruction COS patients and professionals were recruited in a 2:1 ratio so that patients' views were represented preferentially as the procedure is a patient selected optional intervention.[42] In our study, clinical intervention for CES is performed as an emergency so it was deemed appropriate by the study team to have a 1:1 ratio of patients and healthcare professionals. This is to maximise the number of participants involved to help achieve consensus. In addition, the COS should reflect all key stakeholders input equally. If there is an overwhelming response with more than 40 participants interested in attending the consensus meeting the study team will apply stratified purposive sampling. On the day of the consensus meeting, informed consent will be obtained from the patient participants.

Outcomes categorised as 'consensus in' across both stakeholder groups from the Delphi survey (table 4) will be included in the final COS. Outcomes categorised as 'consensus out' across both stakeholder groups from the Delphi survey will be excluded from the final COS. Results of the Delphi survey will be discussed at the consensus meeting, and the main discussion will be regarding the outcomes deemed as achieving 'no consensus' in the

Delphi survey. Participants at the meeting will vote on these outcomes. The same criteria for consensus used in the Delphi survey (table 4) will be used in the consensus meeting. All outcomes that reach 'consensus in' will be included in the COS. All outcomes in the 'consensus out' or 'no consensus' category after voting in the consensus meeting will not be included in the COS. If there is no agreed final COS at the end of the first meeting subsequent meetings will be arranged for this to happen. The participants who had completed both rounds of the Delphi survey would be invited to attend another consensus meeting if required.

## PATIENT INVOLVEMENT

Patients will be involved in the design, review and recruitment of the study. The scope of the research question will be decided with the study team that includes two research partners who are patients with CES. The qualitative interviews will be trailed with the patient research partners, and the topic guide will be reviewed by them. Pilot testing of the Delphi survey will be done by the patient research partners who will be asked to review the patient explanations of the outcomes and the questions on the registration page. Patients will be involved in the recruitment stage of the Delphi as they will be requested via social media to forward the website link for the Delphi survey to any relevant known contacts.

## ETHICS AND DISSEMINATION

We intend to publish the results of the COS for patients with CES in an open access journal. It will also be made available through the CES patient charity websites. Results will be disseminated through international and national presentations. The next step would be to identify the appropriate measurement instrument for each of the outcomes in the COS.[43] COSs are developed in a number of clinical areas and their use is advocated in the UK by the National Institute for Health Research (NIHR) Health Technology Assessment (HTA), Cochrane Reviews of the effects of Healthcare intervention[44] and by WHO handbook for guideline development.[17] The NIHR HTA has added this statement to their application form, "Where established core outcomes exist they should be included among the list of outcomes unless there is a good reason to do otherwise.' By developing the CES COS, we intend to reduce outcome reporting bias, heterogeneity and improve the quality of research studies in CES. This will allow us to synthesise the data and make more robust evidence-based decisions regarding the management of CES.

**Acknowledgements** Special thanks to Ms Claire Thornber and Mr Steven Smith as patient research partners in this study.

**Contributors** NS, MW, SC: conceived the project. TM: is the principal investigator for the study. NS: is the clinical research fellow responsible for management of the project, wrote the protocol and manuscript. TM, PRW, AJN, MW, SC: provide supervision, have input in all aspects of the project, commented on drafts of the manuscript. All authors have read and approved the manuscript.

**Funding** The corresponding author's research fellowship is funded by The Walton Centre Research Funds and The Royal College of Surgeons Research Fellowship.

**Competing interests** None declared.

**Patient consent for publication** Not required.

**Ethics approval** REC and HRA approval was obtained on the 6 December 2016 for the qualitative interviews from South Central—Hampshire A Research Ethic s Committee. REC reference 16/SC/0587. REC and HRA approval was obtained on 26 March 2018 for the Delphi process and consensus meeting from North West—Greater Manchester Central Research Ethic s Committee. REC reference was 18/NW/0022.

**Provenance and peer review** Not commissioned; externally peer reviewed.

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
