## [Reviewer comments · BMJ Open]

ARTICLE DETAILS

TITLE (PROVISIONAL)	PROTOCOL FOR THE DEVELOPMENT OF A CORE OUTCOME SET FOR CAUDA EQUINA SYNDROME: SYSTEMATIC LITERATURE REVIEW, QUALITATIVE INTERVIEWS, DELPHI SURVEY AND CONSENSUS MEETING
AUTHORS	Srikandarajah, Nisaharan; Noble, Adam; Wilby, Martin; Clark, Simon; Williamson, Paula; Marson, Anthony

VERSION 1 - REVIEW

REVIEWER	Jennie Walker Nottingham University Hospitals NHS Trust, United Kingdom
REVIEW RETURNED	17-Jun-2018

GENERAL COMMENTS	The study is one which will be of interest and clinical importance with regard to standardising and directing future research on CES. The study design is appropriate for the intended research outcomes and is ethically sound in its approach. Overall the protocol offers a clear outline of the study, however several amendments are required before publication. The overall tone of the paper is somewhat passive and should be written in the third person as per academic convention. The style of writing is more descriptive than analytical and would benefit from further attention before publication. Several sections require additional detail for clarity. Abstract/introduction – CES can cause significant disability in adults of all ages, not just young adults. The sentence starting ‘the evidence is mainly...’ is fragmented. The term ‘the evidence’ is vague. The aim of the study is identified however the intended benefit is not. Abstract/methods – This section lacks scientific tone with phrases such as ‘a long list’ and ‘all the outcomes mentioned’. This section requires revising to improve clarity and greater focus on the methods used to generate the outcomes set. Abstract/ethics – This should detail who has granted ethical approval rather than just stating the approval has been gained. This will allow readers to identify the appropriateness of the ethical approval. Strengths and limitations of the study – Three of these points lack sufficient critical insight in to the strengths or limitations of the study design and therefore need to be revised. Point 2 is concerned more with method than strength/limitation of the study design.
---

Introduction – contains some sweeping statements which require revising and the addition of supporting references. i.e. ‘CES is most likely due to...’; ‘It is the most common emergency spine operation...’. The incidence of CES would be beneficial to this section as with a summary of the common complications to provide a more comprehensive introduction to the clinical problem.

Rationale for development of COS – ‘through scoping searches’ this is a vague sentence and needs refinement. ‘most patients have had spinal surgery’ is a generalisation and is a fragmented sentence which does not add to the quality or the structure of the section. Avoid using ‘we’ to outline the protocol and rationale. ‘This will be done through a systematic review’ It is not explicit what ‘this’ is.

The protocol does not detail the dates/timeframe for the different phases of the study.

Although the systematic review protocol has been published elsewhere it would be beneficial for the reader is a brief overview of the key points were included here also.

Participant selection – it is unclear what is meant by no discrimination leading to patients going to different clinics. Please clarify what is meant by this.

The term CESI and CESR should be written out in full on the first use and then abbreviated in subsequent usage. Definitions would also aid clarification.

The protocol states that the population will ideally be 50:50 male/female. Please explain what is intended with ‘nesting’ and outline any strategies intended to address male/female participant ratios.

Please clarify patient numbers for patient interviews. The text states up to 10 patients per category, whereas the table states 8-10/10-12. The difference between patient numbers in <2 years and > 2 years should be clearly explained. Please also detail if numbers are capped at this or if additional patients will be interviewed in these categories if one group has a better response rate (until data saturation).

If the patient database has been updated adequately this should remove the risk of contacting patients who have died. This is an important aspect to consider as may cause unnecessary distress to relatives if the database is not checked before contacting patients.

Please identify if participants are sent stamped addressed envelopes to return the response slip – are there alternative options to contact the research team to opt out of the study?

It is convention for the researcher in introduce themselves at provide some background at the start of the interview – please explain/justify why no information will be given to the participant or clarify what is meant by ‘personal information’. It is essential to build participant trust and rapport within the interview, no information about the interviewer may impede the interview. As patient interviews may be emotive a distress protocol is required and should be identified within the study protocol.

	The term 'pressurised' is possibly not the correct term. Poor /unethical interview skills will lead to the patient potentially feeling pressurised, the length of time may result in fatigued patients. The second part of the interview format/analysis section is descriptive rather than offering a considered analytical review/rational of the study protocol and needs revising. The use of a modified Delphi technique should be explained giving the rational for a non-standard approach. Please identify is the CES study team has a patient representative on the panel – this will impact on the decisions made by the team when deciding on inclusion of new outcomes identified by patients who were not involved in the interview stages of the project. Sampling – health care professionals – the term 'a few' needs revising. There are no nursing or AHP representative groups listed and therefore does not reflect the previous section on MDT involvement. It is not clear how long the recruitment phase will last for as the snowball technique will require sufficient time allocated before beginning the second stage. Analysis – Actual parameters need to defined rather than what 'could' be defined as consensus. Consensus meeting. This section requires attention to detail as is currently vague with use of terms such as 'usually', 'most'. Please clarify if the term 'certain organisations' refers to key stakeholder organisations. Please provide details on if participants will receive transportation/ travel expenses/renumeration for time to attend the meetings. It is surprising that 1:1 ratio has been selected for the consensus meeting when the predominant focus to date is identifying patient outcome measures, especially when the protocol highlights similar studies have utilised a 2:1 ratio.
--	---

REVIEWER	Eveline Brouwers Radboud University Medical center, Nijmegen, The Netherlands
REVIEW RETURNED	25-Jun-2018

GENERAL COMMENTS	This will be a interesting research because indeed a little is known about patients with a CES and how they recover after surgery. I'm looking forward to the results. Regarding the study protocol, I think that it is important to visualize the steps you describe in the method section, for instance in a figure. You describe so many steps that you loosing me. From the beginning it is not clear to me what you mean with outcomes and why you want to perform this research. This should be clarified in the abstract and in the introduction. Also the language/word choice is not always correct. Please use some comma's. The reviewer provided a marked copy with additional comments. Please contact the publisher for full details.
--

VERSION 1 – AUTHOR RESPONSE

Reviewer(s)' Comments to Author:

Reviewer: 1

Reviewer Name: Jennie Walker

Institution and Country: Nottingham University Hospitals NHS Trust, United Kingdom

Please state any competing interests or state 'None declared': None declared

Please leave your comments for the authors below

The study is one which will be of interest and clinical importance with regard to standardising and directing future research on CES. The study design is appropriate for the intended research outcomes and is ethically sound in its approach.

Overall the protocol offers a clear outline of the study, however several amendments are required before publication.

The overall tone of the paper is somewhat passive and should be written in the third person as per academic convention. The style of writing is more descriptive than analytical and would benefit from further attention before publication. Several sections require additional detail for clarity.

Abstract/introduction – CES can cause significant disability in adults of all ages, not just young adults. The sentence starting 'the evidence is mainly...' is fragmented. The term 'the evidence' is vague. The aim of the study is identified however the intended benefit is not.

This has been addressed.

Abstract/methods – This section lacks scientific tone with phrases such as 'a long list' and 'all the outcomes mentioned'. This section requires revising to improve clarity and greater focus on the methods used to generate the outcomes set.

This has been addressed.

Abstract/ethics – This should detail who has granted ethical approval rather than just stating the approval has been gained. This will allow readers to identify the appropriateness of the ethical approval.

This has been addressed.

Strengths and limitations of the study – Three of these points lack sufficient critical insight in to the strengths or limitations of the study design and therefore need to be revised. Point 2 is concerned more with method than strength/limitation of the study design.

This has been addressed.

Introduction – contains some sweeping statements which require revising and the addition of supporting references. i.e. 'CES is most likely due to...'; 'It is the most common emergency spine operation...'. The incidence of CES would be beneficial to this section as with a summary of the common complications to provide a more comprehensive introduction to the clinical problem.

This has been addressed.

Rationale for development of COS – ‘through scoping searches’ this is a vague sentence and needs refinement. ‘most patients have had spinal surgery’ is a generalisation and is a fragmented sentence which does not add to the quality or the structure of the section. Avoid using ‘we’ to outline the protocol and rationale. ‘This will be done through a systematic review’ It is not explicit what ‘this’ is.

This has been addressed.

The protocol does not detail the dates/timeframe for the different phases of the study.

Although the systematic review protocol has been published elsewhere it would be beneficial for the reader is a brief overview of the key points were included here also.

This has been addressed.

Participant selection – it is unclear what is meant by no discrimination leading to patients going to different clinics. Please clarify what is meant by this.

This has been removed as was seen as not necessary on review.

The term CESI and CESR should be written out in full on the first use and then abbreviated in subsequent usage. Definitions would also aide clarification.

The protocol states that the population will ideally be 50:50 male/female. Please explain what is intended with ‘nesting’ and outline any strategies intended to address male/female participant rations.

This has been addressed.

Please clarify patient numbers for patient interviews. The text states up to 10 patients per category, whereas the table states 8-10/10-12. The difference between patient numbers in <2 years and > 2 years should be clearly explained. Please also detail if numbers are capped at this or if additional patients will be interviewed in these categories if one group has a better response rate (until data saturation).

This has been addressed. It will be 10 in each category.

If the patient database has been updated adequately this should remove the risk of contacting patients who have died. This is an important aspect to consider as may cause unnecessary distress to relatives if the database is not checked before contacting patients.

This has been addressed.

Please identify if participants are sent stamped addressed envelopes to return the response slip – are there alternative options to contact the research team to opt out of the study?

This has been addressed.

It is convention for the researcher in introduce themselves at provide some background at the start of the interview – please explain/justify why no information will be given to the participant or clarify what is meant by ‘personal information’. It is essential to build participant trust and rapport within the interview, no information about the interviewer may impede the interview.

This has been addressed.

As patient interviews may be emotive a distress protocol is required and should be identified within the study protocol.

The qualitative interviews protocol was agreed by ethics. We had mentioned that the patients GP would be informed regarding the patient's involvement in the study and if there was distress then the GP would be informed. This has been detailed in the protocol.

The term 'pressurised' is possibly not the correct term. Poor /unethical interview skills will lead to the patient potentially feeling pressurised, the length of time may result in fatigued patients.

This has been addressed.

The second part of the interview format/analysis section is descriptive rather than offering a considered analytical review/rational of the study protocol and needs revising.

This has been addressed. Explanation for methodology choice has been placed in the protocol.

The use of a modified Delphi technique should be explained giving the rational for a non-standard approach. Please identify if the CES study team has a patient representative on the panel – this will impact on the decisions made by the team when deciding on inclusion of new outcomes identified by patients who were not involved in the interview stages of the project.

This has been addressed.

Sampling – health care professionals – the term 'a few' needs revising. There are no nursing or AHP representative groups listed and therefore does not reflect the previous section on MDT involvement. It is not clear how long the recruitment phase will last for as the snowball technique will require sufficient time allocated before beginning the second stage.

This has been addressed. Phase 3: The Delphi survey will last for 6 months. There will be a 2-month recruitment period.

Analysis – Actual parameters need to be defined rather than what 'could' be defined as consensus.

This has been corrected.

Consensus meeting. This section requires attention to detail as is currently vague with use of terms such as 'usually', 'most'. Please clarify if the term 'certain organisations' refers to key stakeholder organisations. Please provide details on if participants will receive transportation/ travel expenses/remuneration for time to attend the meetings.

This has been corrected.

It is surprising that 1:1 ratio has been selected for the consensus meeting when the predominant focus to date is identifying patient outcome measures, especially when the protocol highlights similar studies have utilised a 2:1 ratio.

The focus is equal stakeholder input to achieve a consensus amongst patients and healthcare professionals. This is because acute intervention for CES is done as an emergency. This has been mentioned in the protocol.

Reviewer: 2 - please note that this reviewer has also left detailed comments in the manuscript file (please see attachment)

Reviewer Name: Eveline Brouwers

Institution and Country: Radboud University Medical center, Nijmegen, The Netherlands

Please state any competing interests or state 'None declared': nothing to declare

Please leave your comments for the authors below

This will be an interesting research because indeed a little is known about patients with a CES and how they recover after surgery. I'm looking forward to the results. Regarding the study protocol, I think that it is important to visualize the steps you describe in the method section, for instance in a figure. You describe so many steps that you loosing me.

From the beginning it is not clear to me what you mean with outcomes and why you want to perform this research. This should be clarified in the abstract and in the introduction. Also, the language/word choice is not always correct. Please use some comma's.

The comments here and on the reviewers PDF have been addressed. Figure 1 gives an outline of the study.

VERSION 2 – REVIEW

REVIEWER	Eveline Brouwers Radboud UMC, Nijmegen, The Netherlands
REVIEW RETURNED	15-Nov-2018

GENERAL COMMENTS	This manuscript is much more clear and understandable compared with the first version.
--